# MLP Embedded Inverse Tone Mapping

## ABSTRACT

The advent of High Dynamic Range/Wide Color Gamut (HDR/WCG) display technology has made significant progress in providing exceptional richness and vibrancy for the human visual experience. However, the widespread adoption of HDR/WCG images is hindered by their substantial storage requirements, imposing significant bandwidth challenges during distribution. Besides, HDR/WCG images are often tone-mapped into Standard Dynamic Range (SDR) versions for compatibility, necessitating the usage of inverse Tone Mapping (iTM) techniques to reconstruct their original representation. In this work, we propose a meta-transfer learning framework for practical HDR/WCG media transmission by embedding image-wise metadata into their SDR counterparts for later iTM reconstruction. Specifically, we devise a meta-learning strategy to pre-train a lightweight multilayer perceptron (MLP) model that maps SDR pixels to HDR/WCG ones on an external dataset, resulting in a domain-wise iTM model. Subsequently, for the transfer learning process of each HDR/WCG image, we present a spatial-aware online mining mechanism to select challenging training pairs to adapt the meta-trained model to an image-wise iTM model. Finally, the adapted MLP, embedded as metadata, is transmitted alongside the SDR image, facilitating the reconstruction of the original image on HDR/WCG displays. We conduct extensive experiments and evaluate the proposed framework with diverse metrics. Compared with existing solutions, our framework shows superior performance in fidelity (up to 3dB gain in perceptual-uniform PSNR), minimal latency (1.2s for adaptation and 2ms for reconstruction of a 4K image), and negligible overhead (40KB).

## CCS CONCEPTS

• **Computing methodologies** → **Reconstruction**.

## KEYWORDS

Inverse Tone Mapping, High Dynamic Range, Wide Color Gamut

## 1 INTRODUCTION

The concept of High Dynamic Range/Wide Color Gamut (HDR/WCG) media, as defined in literature [32, 36], encompasses an extended range of luminance representation and an augmented capability for reproducing a broader spectrum of visible colors, significantly surpassing the limitations of the Standard Dynamic Range (SDR) media definition. The advent of HDR/WCG display technology represents

**Unpublished working draft. Not for distribution.**

Permission to make digital or hard copies of all or part of this work for personal or classroom use is granted without fee provided that copies are not made or distributed for profit or commercial advantage and that copies bear this notice and the full citation on the first page. Copyrights for components of this work owned by others than the author(s) must be honored. Abstracting with credit is permitted. To copy otherwise, or republish, to post on servers or to redistribute to lists, requires prior specific permission and/or a fee. Request permissions from permissions@acm.org.

*ACM MM, 2024, Melbourne, Australia*

© 2024 Copyright held by the owner/author(s). Publication rights licensed to ACM.
ACM ISBN 978-x-xxxx-xxxx-x/YY/MM
https://doi.org/10.1145/nnnnnnn.nnnnnnn

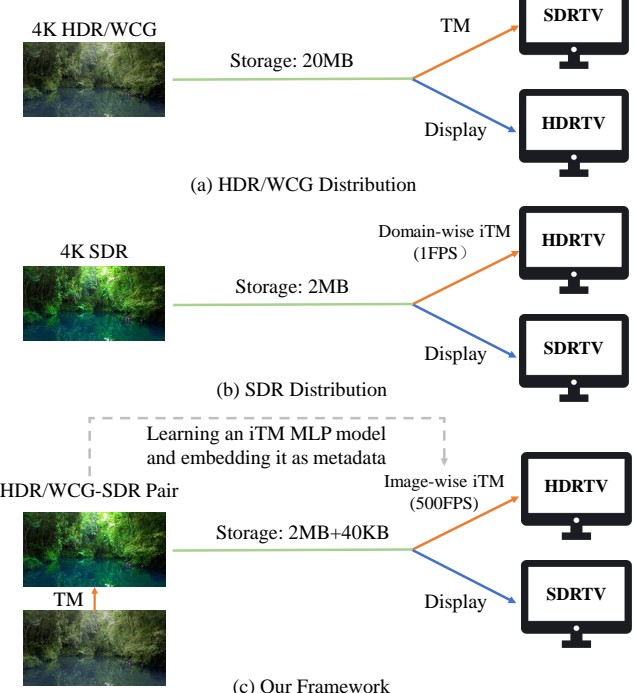

**Figure 1: HDR/WCG media compatibility solutions. a) Transmission in original quality, and applying Tone Mapping (TM) to HDR/WCG content for SDRTV compatibility. b) Transmission in SDR and employing inverse Tone Map (iTM) for HDRTV compatibility. c) Our framework: Learning an MLP, embedding it as metadata, and conducting image-wise iTM with the embedded MLP.**

a noteworthy milestone in the multimedia sector, offering unparalleled richness and vibrancy in visual experiences. However, the widespread adoption of HDR/WCG technology encounters substantial challenges, primarily due to its extensive storage and bandwidth requirements during distribution [32]. For instance, as depicted in Fig. 1(a), a typical 4K HDR/WCG media requires 20MB for storage and a bandwidth of 25Mbps for real-time streaming.

At present, the concurrent usage of HDR/WCG and SDR media remains the prevailing practice. On one hand, owing to the prevalence of SDR displays, HDR/WCG contents are frequently tone-mapped (TM) to their SDR counterparts to ensure compatibility [11, 34]. On the other hand, there exists a considerable amount of SDR content that requires conversion into HDR/WCG versions for display on HDR/WCG screens [33]. Consequently, there is a substantial demand for inverse Tone Mapping (iTM) methods that aim to predict lost HDR/WCG information from SDR versions, as illustrated in Fig. 1(b).

Existing iTM methods encounter several significant challenges. Rule-based iTM methods often suffer from artifacts and inadequate detail reconstruction, particularly in poorly exposed areas [14, 34]. Learning-based methods, on the other hand, rely on an external dataset to model the iTM task as a mapping problem between SDR and HDR/WCG domains [3, 4, 7, 9, 12–14, 42]. However, these methods usually assume a predefined TM operator, which may not consistently align with SDR inputs from various sources. Furthermore, due to the inherently ill-posed nature of domain-wise iTM, these methods often require significant computational resources, thereby failing to meet the requirements for real-time presentation of high-definition HDR/WCG content.

In this work, as shown in Fig. 1(c), we present a novel meta-transfer framework for the practical transmission of HDR/WCG media by embedding a learned multilayer perceptron (MLP) model as metadata into their SDR versions, where the embedded MLP is employed for image-wise iTM reconstruction. Specifically, our framework contains three stages, *i.e.*, meta-training, transfer, and reconstruction stages. Firstly, in the meta-training stage, we take advantage of an external dataset to train a lightweight MLP model that maps SDR pixels to HDR/WCG ones. This MLP serves as a domain-wise iTM model, and we devise a meta-learning strategy to prepare it as an initialization point for fast adaptation to an image-wise one. Secondly, in the transfer stage, we utilize pixel samples from the current HDR/WCG-SDR pair to fine-tune the meta-trained MLP and embed it as metadata in the SDR image. We adopt an online hard example mining (OHEM) mechanism to select training pixel pairs according to spatial-aware feedback from the loss function, improving the adaptation performance to the current image. Finally, in the reconstruction stage, we restore the HDR/WCG content from the SDR version with the embedded MLP model for image-wise iTM.

We evaluate the proposed framework with diverse metrics, including perceptual uniform, HDR-tailored, chromaticity, and conventional ones. Compared with existing solutions, our framework shows superior performance in fidelity (up to 3dB gain in perceptual-uniform PSNR), minimal latency (1.2s for adaptation and 2ms for reconstruction of a 4K image), and negligible overhead (40KB). Besides, we show that our framework enjoys better generalization capacity to different TM operators and an application of our framework to accelerate domain-wise iTM methods via distillation.

To summarize, our contributions are three-fold:

- We propose a meta-transfer learning framework for HDR/WCG media transmission by embedding a learned image-wise iTM MLP as metadata to SDR versions.
- We devise a meta-learning strategy and a spatial-aware online mining mechanism to improve the generalization and adaptation capacity of the iTM model.
- Compared to existing methods, our framework exhibits superior performance, lower latency, and less overhead, showing its effectiveness and practicability.

## 2 RELATED WORKS

### 2.1 Inverse Tone Mapping

The objective of iTM is to transform SDR images into HDR/WCG images. The traditional iTM methods mainly focus on adjusting the parameters of the global mapping function based on the image content [1, 21, 27] and employ edge enhancement in saturated regions [15]. Recently, learning-based iTM boasts superior performance compared to traditional methods. Initially, researchers [12–14] attempt to address both iTM and super-resolution jointly. Chen et al. [3] are the pioneers in treating the iTM task as an independent task and utilizing Unet [29] for end-to-end iTM. Chen et al. [4] propose HDRTVNet which consists of global color mapping, local enhancement, and highlight generation stages. Recently, Xu et al. [42] propose FMNet that incorporates frequency analysis to improve the reconstruction performance. Huang et al. [9] propose ICtCpNet to further improve the iTM performance by separating the chromaticity and luminance channels. Guo et al. [7] propose a segmentation-based solution with various TM operators. While existing methods have demonstrated promising performance, they often assume a predefined TM operator and model the iTM task as a domain-wise mapping problem. In this work, we introduce the idea of embedding metadata to SDR images and extend the concept of iTM from domain-wise level (pre-defined TM operator) to image-wise level (both TM operator and image content).

### 2.2 Metadata Based Reconstruction

Recently, the idea of embedding metadata into low-quality data (*e.g.*, sRGB image) for later reconstruction of high-quality data (*e.g.*, RAW image) has been explored in other tasks, *i.e.*, RAW image reconstruction and color gamut expansion. Rang and Brown [26] store a set of RAW reconstruction parameters, which model the typical operations in image processing pipeline. Punnappurath and Brown [25] present an algorithm that estimates interpolation parameters in reconstruction stage, utilizing uniformly sampled RAW image pixels stored as metadata. Nam et al. [22] propose a sampling strategy that selects representative RAW pixels based on superpixels using a pre-trained Unet [29]. Another Unet is trained to reconstruct RAW images from sRGB images and sampled pixels. Li et al. [17] attempt to employ an implicit neural function to map coordinates to RAW pixels conditioned with sRGB pixels. Beyond RAW reconstruction, metadata-based techniques are also applied in the domain of color gamut expansion, as demonstrated by GamutMLP [16]. In this method, pixels in WCG (BT.2020 [32]) but out of standard color gamut (BT.709 [2]) are randomly sampled to train an MLP. The fine-tuned weights are embedded as metadata to assist in the reconstruction of WCG images. We introduce this idea into the iTM task. Further, we propose a meta-learning pre-training strategy and spatial-aware OHEM mechanism to further improve its performance and efficiency for image-wise iTM.

### 2.3 Meta Learning

Meta-learning, also known as learning to learn, facilitates rapid adaptation to new tasks. It is widely integrated in various computer vision tasks, *e.g.*, classification [28, 30], detection [39], segmentation [18], and tracking [38]. Finn et al. [6] initialize model parameters and adapt them to support sets in the inner loop. Then, they use these adapted parameters to measure the loss in query sets and finally update the original parameters with this loss, namely the outer loop. Nichol et al. [23] simplify meta-learning by directly updating meta-network parameters using the difference between

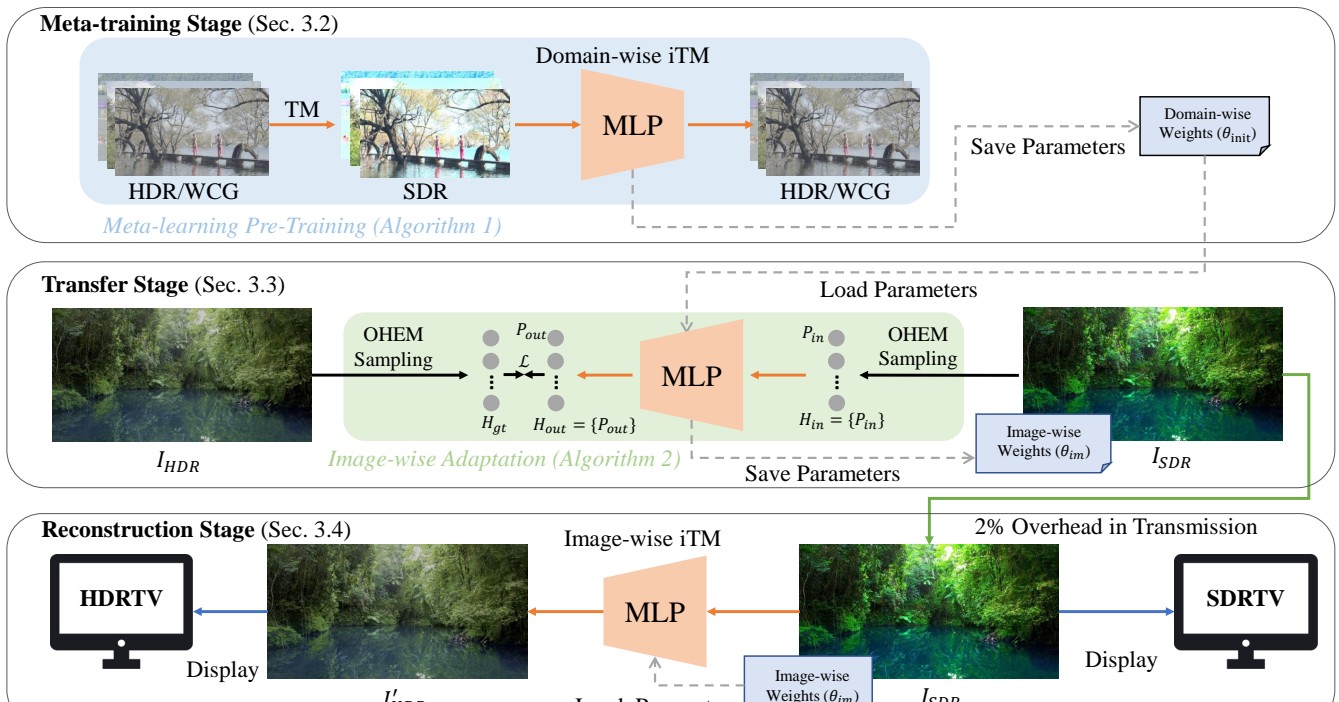

**Figure 2: Overview of our proposed framework. 1) Meta-training stage: We leverage an external HDR/WCG-SDR dataset to pre-train a lightweight MLP that maps SDR pixels to HDR/WCG ones with a meta-learning strategy, obtaining the pre-trained weights $\theta_{init}$. 2) Transfer stage: We sample a portion of pixels with the proposed spatial-aware OHEM mechanism from the HDR/WCG image $I_{HDR}$ and the SDR image $I_{SDR}$ to construct pixels sets $H_{in}$ and $H_{gt}$, respectively. Then, $H_{in}$ and $H_{gt}$ are utilized to fine-tune the MLP model $M_{\theta_{init}}$, which is initialized with $\theta_{init}$, yielding the image-wise weights $\theta_{im}$. $\theta_{im}$ is then embedded into $I_{SDR}$ as metadata. 3) Reconstruction stage: For HDRTV compatibility, the HDR/WCG representation $I'_{HDR}$ can be reconstructed with the embedded MLP from $I_{SDR}$.**

original and inner updated parameters. In our framework, we treat each image's iTM task as an individual task. Different from the pre-training strategy in GamutMLP, we introduce a fast-adaptation step before the outer loop, integrating the procedure prior to the transfer stage into our meta-training stage.

### 2.4 Online Hard Example Mining

OHEM [35] is a methodology aimed at enhancing model performance through the selective inclusion of challenging samples. In contrast to conventional uniform sampling mechanisms, OHEM prioritizes samples that significantly contribute to enhancing model performance. Shrivastava et al. [35] first apply it in the object detection domain. After that, OHEM is widely integrated with classification tasks [5, 8]. In our framework, we adapt OHEM by introducing spatial-aware feedback from pixel-level error maps.

## 3 MLP EMBEDDED INVERSE TONE MAPPING

### 3.1 Framework Overview

As shown in Fig. 2, our framework contains three principal stages: the meta-training stage, the transfer stage, and the reconstruction stage. During the meta-training stage, we leverage an external

HDR/WCG dataset and randomly sample HDR/WCG images, which are used to generate data pairs via a predefined TM operator. These data pairs are subsequently employed in a meta-learning manner to pre-train a lightweight MLP that maps SDR pixels to HDR/WCG pixels, resulting in pre-trained domain-wise weights $\theta_{init}$.

In the transfer stage, taking advantage of pre-trained $\theta_{init}$, the initialized MLP $M_{\theta_{init}}$ is enabled to be fine-tuned from a universal MLP to an image-wise one easily. For each HDR/WCG image $I_{HDR}$ and SDR image $I_{SDR}$ pair, we sample a small portion of pixels to construct the pixel sets $H_{gt}$ and $H_{in}$, respectively. $H_{in}$ is then fed into $M_{\theta_{init}}$, yielding with prediction pixel set $H_{out}$. Finally, the loss $\mathcal{L}$ between $H_{gt}$ and $H_{out}$ is used to fine-tune $M_{\theta_{init}}$. What's more, we adopt an OHEM mechanism to select challenging training pixel pairs according to spatial-aware feedback from the loss function, improving the adaptation performance to the current image. After adaptation on the current HDR/WCG-SDR pair, we get the image-wise weights $\theta_{im}$ and embed them into $I_{SDR}$ as metadata.

As to the reconstruction stage, for HDRTV compatibility, we input $I_{SDR}$ into the MLP, which is loaded with image-wise weights $\theta_{im}$ embedded in the $I_{SDR}$. Consequently, the 4K HDR/WCG image $I'_{HDR}$ can be reconstructed with high efficiency.

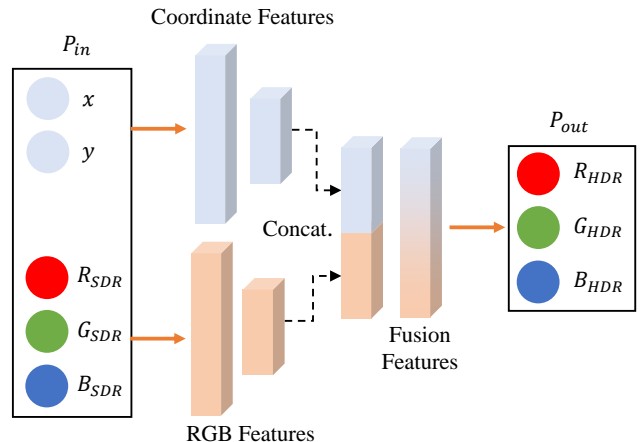

Coordinate Features

$P_{in}$

$x$

$y$

$R_{SDR}$

$G_{SDR}$

$B_{SDR}$

RGB Features

Concat.

Fusion Features

$P_{out}$

$R_{HDR}$

$G_{HDR}$

$B_{HDR}$

Figure 3: The network structure of the MLP. Inspired by INF [17], we adopt simplified version of its MLP architecture where the implicit neural function mapping coordinate $(x, y)$ to pixel $(R_{HDR}, G_{HDR}, B_{HDR})$ is conditioned by SDR pixel $(R_{SDR}, G_{SDR}, B_{SDR})$.

---

**Algorithm 1** Meta-learning pre-training.

**Require:** external HDR/WCG-SDR pairs, max epoch $N$.

Randomly initialize an MLP with $\theta_0$

$n \leftarrow 0$

**while** $n < N$ **do**

    Sample a support set $S_n$ and a query set $Q_n$

    $\mathcal{L}_{S_n} = \sum_{(I_{SDR}, I_{HDR}) \in S_n} \mathcal{D}_{MSE}(I_{HDR}, M_{\theta_n}(I_{SDR}))$

    Inner update: $\theta'_n = \theta_n + \mathcal{L}_{S_n}$

    Initialize an image-wise MLP $M$ with $\theta'_n$

    **while** not converged **do**

        $\mathcal{L}_{Q_n} = \sum_{(I_{SDR}, I_{HDR}) \in Q_n} \mathcal{D}_{MSE}(I_{HDR}, M_{\theta'_n}(I_{SDR}))$

        $\theta'_n = \theta'_n + \mathcal{L}_{Q_n}$

    **end while**

    $\mathcal{L}_{meta} = \sum_{(I_{SDR}, I_{HDR}) \in Q_n} \mathcal{D}_{MSE}(I_{HDR}, M_{\theta'_n}(I_{SDR}))$

    Outer update: $\theta_{n+1} = \theta_n + \mathcal{L}_{meta}$

    $n \leftarrow n + 1$

**end while**

Save the meta-trained MLP model $M_{\theta_{init}}$

---

## 3.2 Pre-training with Meta-Learning

Inspired by INF [17], we adopt an MLP architecture where the implicit neural function mapping coordinate $(x, y)$ to pixel $(R_{HDR}, G_{HDR}, B_{HDR})$ is conditioned by the input SDR pixel $(R_{SDR}, G_{SDR}, B_{SDR})$. As shown in Fig. 3, we input the coordinate and SDR pixel into two branches respectively, resulting in coordinate features and RGB features. Then all the features are concatenated up to fusion features which finally output HDR pixels.

In order to make our framework more adaptive and maximize the benefits of image-wise adaptation, we propose a meta-learning pre-training strategy, as shown in Fig. 4. For each iteration, we construct a task by randomly sampling a support set $S_n$ and a query set $Q_n$ from external HDR/WCG-SDR data pairs. Initially, our focus

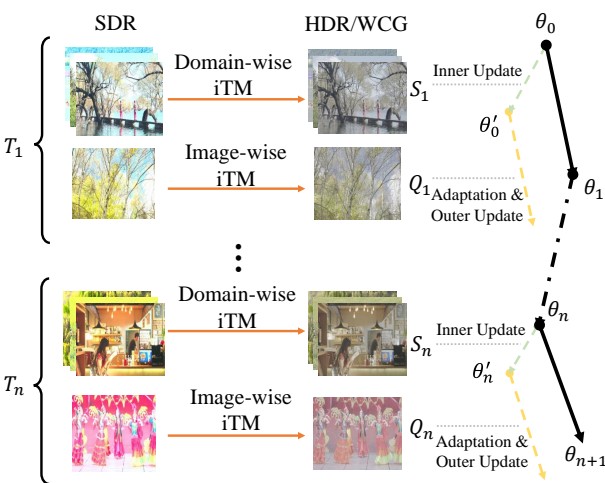

SDR     HDR/WCG     $\theta_0$

$T_1$

Domain-wise iTM    Inner Update    $S_1$    $\theta'_0$

Image-wise iTM    $Q_1$   Adaptation & Outer Update    $\theta_1$

$T_n$

Domain-wise iTM    Inner Update    $S_n$    $\theta_n$

   $\theta'_n$

Image-wise iTM    $Q_n$   Adaptation & Outer Update    $\theta_{n+1}$

Figure 4: Meta-learning Pre-training. We construct a task by randomly sampling a support set $S_n$ and a query set $Q_n$ from external HDR/WCG-SDR data pairs. In each iteration, the MLP is firstly trained on $S_n$, obtaining an inner-updated model with parameter $\theta'_n$. Then, we adapt this model to $Q_n$ and conduct outer updating.

lies on facilitating the MLP's acquisition of a universal domain mapping capability from SDR pixels to HDR/WCG ones, termed a domain-wise iTM task. So we update $\theta_n$ to $\theta'_n$ via $S_n$, namely the inner update. In the common meta-learning paradigm [6], $\theta_n$ is updated $\theta_{n+1}$ directly with $Q_n$ and $\theta'_n$, i.e., the outer update. But, in our framework, we aim at image-wise iTM, where the iTM model is adapted to the HDR/WCG-SDR image pair before inference. Thus, we introduce a fast-adaptation step before the outer update, integrating the procedure prior to the transfer stage into our meta-train stage. In this step, we adapt $\theta'_n$ to $Q_n$ and then we update the $\theta_n$ to $\theta_{n+1}$, via the reconstruction loss calculated by applying the MLP with $\theta'_n$ on $Q_n$. Finally, we can get a meta-trained MLP model $M_{\theta_{init}}$, which can be easily adapted from a domain-wise iTM model to an image-wise one. We summarize this strategy in Algorithm 1.

## 3.3 Transfer with Spatial-aware OHEM

After getting the initialization weights $\theta_{init}$ and loading it into $M_{\theta_{init}}$, for each HDR/WCG-SDR image pair, we fine-tune a domain-wise MLP to an image-wise one. In detail, as shown in Fig. 2, we sample pixels from the HDR/WCG image $I_{HDR}$ and the SDR image $I_{SDR}$ to construct pixel sets $H_{in}$ and $H_{gt}$. Then $H_{in}$ and $H_{gt}$ are utilized to fine-tune $M_{\theta_{init}}$, yielding the image-wise weights $\theta_{im}$, which are embedded into $I_{SDR}$ as metadata. Previous metadata-based reconstruction methods, either uniform [17, 25] or non-uniform [16, 22], only utilize a very limited portion of the HDR pixels (2%) in the sampling process. This limitation leads to poor reconstruction quality in hard examples, i.e., saturated regions (as shown later in Fig. 8). Thus, inspired by the seminal work of OHEM [35], we propose the spatial-aware OHEM mechanism, as shown in Fig. 5. For a certain interval during fine-tuning, we get spatial-aware feedback via

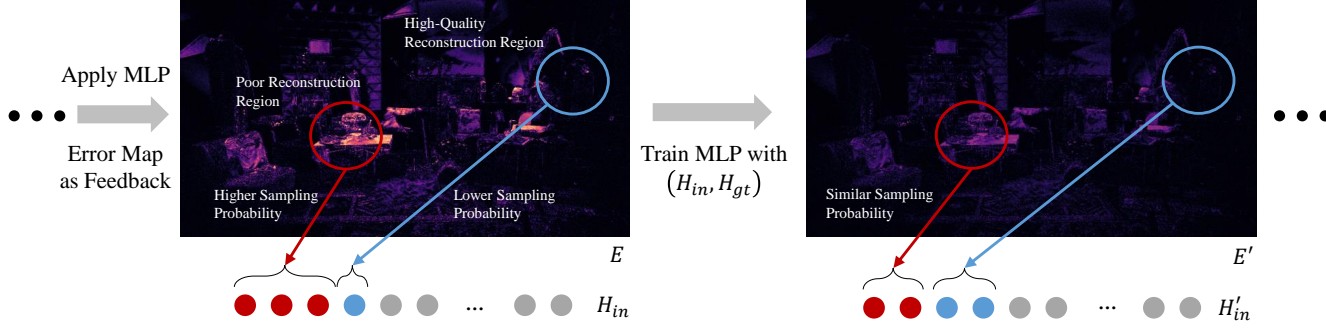

**Figure 5: Spatial-aware OHEM mechanism. For a certain interval during fine-tuning, we get spatial-aware feedback by applying the MLP model and obtaining an error map $E$, indicating the current reconstruction performance for each pixel. Then, we sample HDR/WCG-SDR pixels ($H_{gt}$ and $H_{in}$) according to $E$. This way, the following training process is more focused on poorly reconstructed regions.**

---

**Algorithm 2** Image-wise adaptation.

---

**Require:** SDR image $I_{SDR}$, HDR/WCG image $I_{HDR}$, meta-trained MLP model $M_{\theta_{init}}$
  **while** not converged **do**
    **if** Resample **then**
      $E = \mathcal{D}_{L_1}(M_{\theta_{init}}(I_{SDR}), I_{HDR})$
      Sample $H_{in}$ and $H_{gt}$ from $I_{SDR}$ and $I_{HDR}$ based on $E$
    **end if**
    $H_{out} \leftarrow \{M_{\theta_{init}}(P_{in})\}\ for\ P_{in} \in H_{in}$
    $\mathcal{L} = \mathcal{D}_{MSE}(H_{out}, H_{gt})$
    update MLP parameter $\theta$ with $\mathcal{L}$
  **end while**
  Save the adapted MLP model $M_{\theta_{im}}$

---

applying the MLP model and obtaining an error map $E$, which indicates the current reconstruction performance of each pixel. Then we prioritize selecting pixels with larger errors, *e.g.*, the pixels in the red circle of Fig. 5. Such mechanism make our fine-tuning process more focused on poorly reconstructed regions. Our spatial-aware OHEM mechanism is summarized as Algorithm 2.

### 3.4 Reconstruction with Embedded MLP

As shown in Fig. 2, the image-wise weights $\theta_{im}$ are embedded into SDR image $I_{SDR}$ for transmission and distribution. Our framework is naturally compatible with SDR displays. For HDRTV compatibility, we input $I_{SDR}$ into the MLP, which is loaded with $\theta_{im}$. Hence, our framework not only ensures compatibility with SDRTV but also excels in the high-quality and efficient reconstruction of HDR/WCG content for seamless integration with HDRTV, all achieved within the constraints of a mere 2% increase in data.

## 4 EXPERIMENTS

### 4.1 Settings

**Datasets**. We utilize the HDRTV4K dataset [7] to evaluate the effectiveness of our proposed framework. This dataset consists of 3,878 BT.2020/PQ1000 4K HDR images. Following previous works [4, 7,

42], we adopt YouTube TM operator to generate data pairs. We split them according to the official split [7] into a training set and a test set, which includes 3,478 and 400 images, respectively.

**Evaluation metrics**. Conventional image quality metrics, such as PSNR and SSIM [40], can not be directly computed on linear high dynamic range color values because such values are non-linearly related to the perception of visible differences [19]. Instead, in our experiments, we adopt the perceptual uniform (PU) encoder [19] to convert the absolute values of HDR/WCG images into PU values. In this PU space, we employ metrics, including PSNR, PSNR (Y), MS-SSIM [41], feature similarity (FSIM) [44], and visual saliency-induced index (VSI) [43], to evaluate the performance of our framework and comparison methods. Additionally, we report the HDR-VDP3 [20] metric that is tailored for HDR/WCG image quality assessment. However, most of the metrics mentioned above operate in the luminance domain (HDR), so we introduce the chromaticity metric $\Delta_{ITP}$ [10] to evaluate the reconstruction results in terms of chromaticity (WCG). Finally, we also report the conventional metrics, PSNR and SSIM[40], as a reference. We report running latency and model size for efficiency evaluation.

**Comparison methods**. We compare our framework with several state-of-the-art domain-wise iTM methods, including HDRTVNet [4], FMNet [42], and ICtCpNet [9]. Moreover, we include metadata-based reconstruction methods designed for other tasks, including GamutMLP [16] for color gamut expansion, CAM [22], and INF [17] for RAW reconstruction.

**Implementation details**. Our MLP consists of four layers and the channel numbers are 64, 32, 64, and 3. In the meta-training stage, each task consists of 3 data pairs in the support set and 1 data pair in the query set. For each task, we fine-tune the model for 500 epochs at a learning rate of $1 \times 10^{-4}$. In the transfer stage, we resample the pixel sets in the last 25 percent of the adaptation epochs at an interval of 50 epochs. We get the official codes of domain-wise iTM methods and fully retrain their models. As to the metadata-based reconstruction methods, we make necessary modifications based on the official codes to adapt them to the iTM task. Our code is based on PyTorch [24] and will be made publicly available upon acceptance.

| Method | Perceptual Uniform | | | | | HDR | Chromaticity | Conventional | |
|---|---|---|---|---|---|---|---|---|---|
| | PSNR↑ | PSNR (Y)↑ | MS-SSIM↑ | FSIM↑ | VSI↑ | HDR-VDP3↑ | $\Delta E_{ITP}$↓ | PSNR↑ | SSIM↑ |
| HDRTVNet [4] | 31.20 | 32.31 | 0.9738 | 0.9766 | 0.9899 | 8.798 | 8.493 | 35.57 | 0.9539 |
| FMNet [42] | 32.13 | 33.16 | 0.9701 | 0.9719 | 0.9889 | 8.633 | 6.550 | 36.11 | 0.9552 |
| ICTCPNet [9] | 33.12 | 34.46 | 0.9770 | 0.9783 | 0.9909 | 8.895 | 6.749 | 36.75 | 0.9565 |
| GamutMLP [16] | 34.44 | 36.26 | 0.9754 | 0.9811 | 0.9961 | 9.041 | 9.680 | 35.23 | 0.8841 |
| INF [17] | 36.93 | 37.86 | 0.9825 | 0.9886 | 0.9958 | 9.340 | 4.295 | 40.55 | 0.9660 |
| Ours | **38.62** | **39.62** | **0.9900** | **0.9900** | **0.9967** | **9.463** | **4.005** | **41.77** | **0.9689** |

**Table 1: Quantitative comparsion on the HDRTV4K dataset [7]. The perceptual uniform metrics are obtained with the PU encoding [19]. The conventional metrics are evaluated directly within the PQ-encoded HDR/WCG space.**

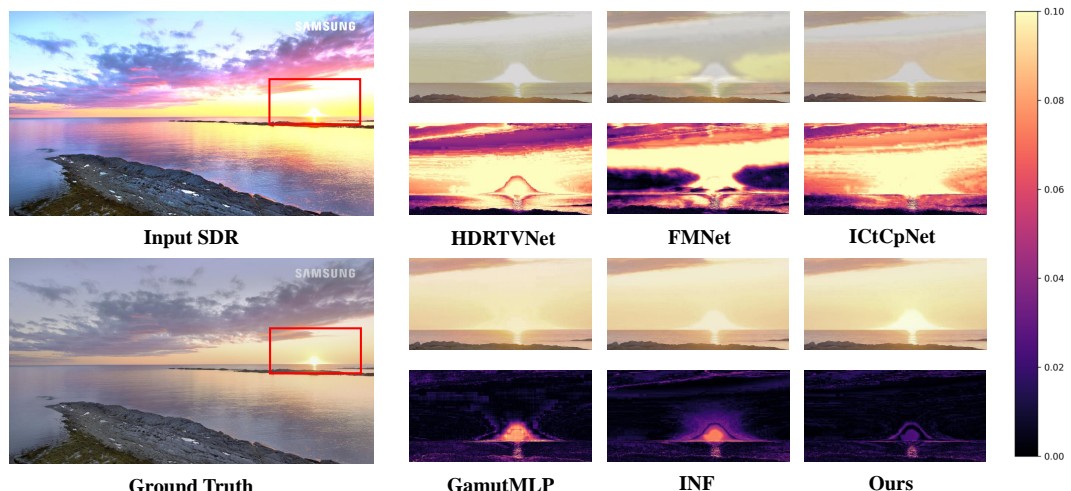

**Figure 6: Qualitative comparison between the input SDR, Ground Truth, and results of HDRTVNet [4], FMNet [42], ICtCpNet [9], GamutMLP [16], INF [17] and our framework. We visualize the error maps below the results.**

| Method | Latency | | Extra Data |
|---|---|---|---|
| | Transfer | Reconstruction | |
| HDRTVNet [4] | - | 3752.00ms | - |
| FMNet [42] | - | 724.10ms | - |
| ICTCPNet [9] | - | 3136.00ms | - |
| GamutMLP [16] | 8.58s | 70.80ms | 23KB |
| INF [17] | 13.46s | 4.13ms | 2.9MB |
| Ours | 1.22s | 2.13ms | 40KB |

**Table 2: Efficiency results on an NVIDIA RTX 4090 GPU.**

## 4.2 Performance Evaluation

**Quantitative results.** Table 1 summarizes the performance of our framework and comparison methods. Our framework has significant improvement in all metrics, even compared with the most competitive domain-wise iTM method ICtCpNet (38.62dB vs. 33.12dB in PU PSNR) and metadata-based method INF (38.62 dB vs. 36.93 dB in PU PSNR). What's more, we should note that GamutMLP performs

quite well in the HDR-VDP3 and PSNR (Y) metrics, but performs poorly in the $\Delta_{ITP}$, which indicates the quality of chromaticity accuracy. We attribute this phenomenon to that its sampling strategy where most of the training pairs are sampled from out-of-gamut regions is not suitable for the iTM task, which requires the algorithm to model the luminance and chromaticity mapping relationship between SDR and HDR/WCG simultaneously. In addition, because CAM [22] consumes more than 64GB of GPU memory for 4K resolution, we compare our framework with CAM on a 1080p downsampled version of the HDRTV4K dataset. Our framework achieves 37.88dB, 9.529, 4.467 in PU PSNR, HDR-VDP3, and $\Delta_{ITP}$, while those of CAM are 34.25dB, 9.266, and 6.639.

**Qualitative results.** We show reconstructed HDR/WCG images and error maps in Fig. 6. Existing domain-wise methods suffer from a lack of highlighted pixels and color shifts, while metadata-based methods, especially our framework, could reconstruct the overexposure regions accurately. In Fig. 7, for the input SDR image, the ground truth HDR/WCG image, and reconstruction results, the pixel distributions are visualized in the CIE xyY space [31]. Compared to HDR/WCG images, SDR images suffer significant loss of

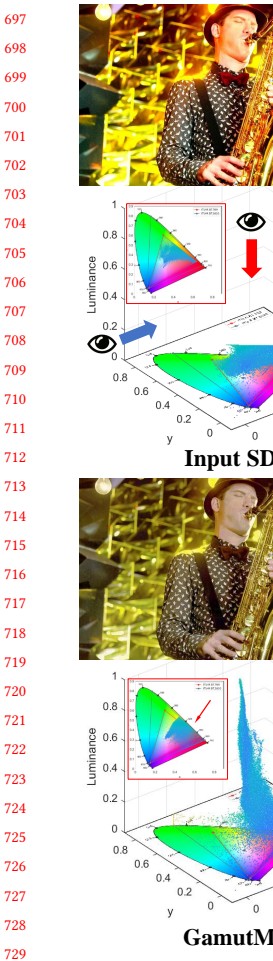
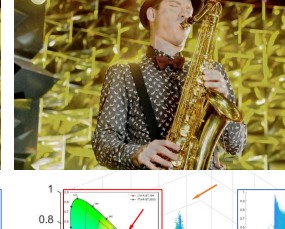
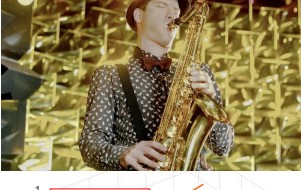
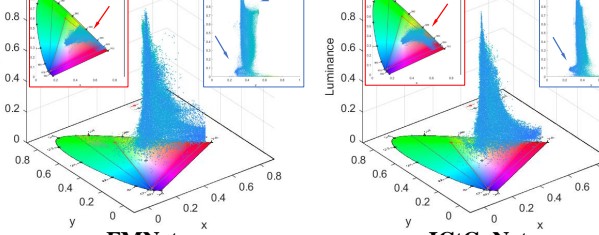
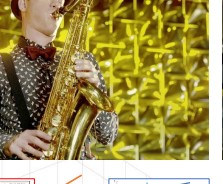
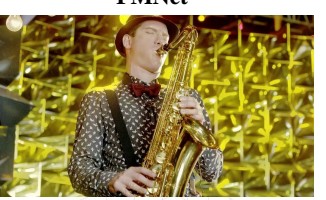
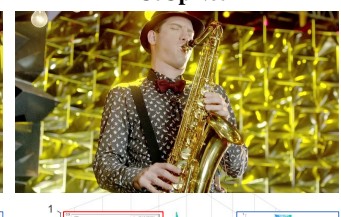
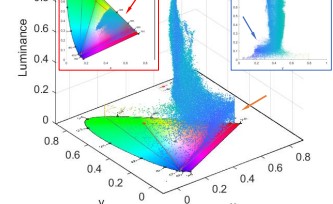
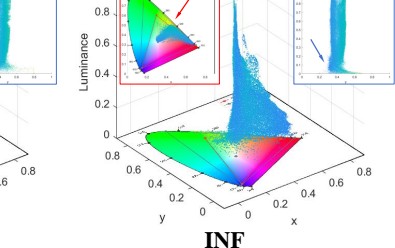
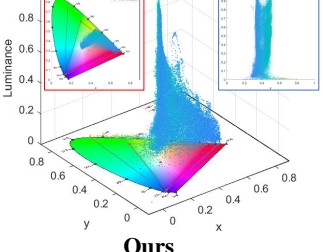
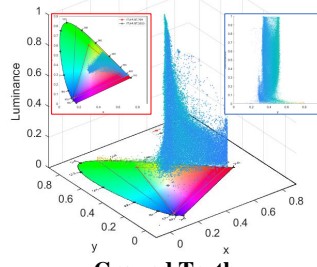

**Figure 7: Qualitative comparison between input SDR, ground truth, and results of HDRTVNet [4], FMNet [42], ICtCpNet [9], GamutMLP [16], INF [17] and our framework. The pixel distributions are visualized in the CIE xyY space [31]. Illustrations within the red depict the distribution in terms of chromaticity (xy), while those within the blue in terms of luminance (Y).**

information in both the luminance and color gamut. HDRTVNet, FMNET, and ICtCpNet have demonstrated notable performance in chromaticity domain reconstruction. However, in terms of luminance domain reconstruction, these methods exhibit shortcomings in reconstructing high-luminance pixels adequately. GamutMLP suffers from inaccurate chromaticity domain reconstruction, while INF exhibits relative weaknesses in luminance domain. More visual comparisons are available in supplementary materials.

### 4.3 Efficiency Evaluation

In practical HDR/WCG media transmission, latency and storage consumption are also essential metrics. As listed in Table 2, we evaluate the efficiency of our framework. Compared with domain-wise iTM methods, our framework only needs 2.13ms to reconstruct a 4K resolution image, while FMNet needs 724.10ms. Our advantage will be even more pronounced, when compared with metadata-based methods, like INF (1.22s vs. 13.46s in transfer time and 40KB vs. 2.9MB in extra data). Our framework has a comparable scale in

the size of extra data compared to GamutMLP (40KB vs. 23KB) and exhibits a significant superiority of over 30 times faster in terms of reconstruction latency (2.13ms vs. 70.8ms).

### 4.4 Generalization Evaluation

As shown in Table 3, when the inference (OCIO [37]) TM operator is different from training (YouTube), domain-wise iTM methods perform poorly, showing the limitation of domain-wise iTM methods. In comparison, metadata-based methods can reconstruct the HDR/WCG images accurately. Compared with other metadata-based methods, our framework still has a significant lead in metrics (especially in PU PSNR and $\Delta_{ITP}$), demonstrating the superior generalization capacity of our framework.

### 4.5 Ablation Analysis

**The effectiveness of the meta-learning strategy.** We evaluate the performance of our framework with the naive pre-training

| Methods | PSNR↑ | HDR-VDP3↑ | $\Delta_{ITP}$↓ |
|---|---|---|---|
| HDRTVNet [4] | 19.54 | 8.158 | 32.89 |
| FMNet [42] | 19.79 | 8.088 | 31.65 |
| ICtCpNet [9] | 19.95 | 8.231 | 31.35 |
| GamutMLP [16] | 36.23 | 9.354 | 7.816 |
| INF [17] | 38.12 | 9.637 | 5.003 |
| Ours(w/o meta learning) | 38.57 | 9.600 | 5.883 |
| Ours | 39.42 | 9.684 | 4.448 |

Table 3: Generalization evaluation results. Methods are trained with YouTube TM operator, but tested with OCIO [37]. The PNSR value is evaluated in PU [19] encoded domain.

| Pre-training | Sampling | PSNR↑ | HDR-VDP3↑ |
|---|---|---|---|
| No Pre-train | Uniform | 33.70 | 9.034 |
| Naive Pre-train | Uniform | 37.22 | 9.237 |
| Naive Pre-train | Spatial-aware OHEM | 38.30 | 9.430 |
| GamutMLP Pre-train | Uniform | 37.90 | 9.423 |
| GamutMLP Pre-train | Spatial-aware OHEM | 38.15 | 9.432 |
| Meta Learning | Uniform | 38.47 | 9.457 |
| Meta Learning | Spatial-aware OHEM | 38.62 | 9.463 |

Table 4: Ablation results of the proposed meta-learning pre-training strategy and spatial-aware OHEM mechanism. The PNSR value is evaluated in PU [19] encoded domain.

**Naive Pre-train without OHEM** · **Naive Pre-train with OHEM**

**Ours without OHEM** · **Ours with OHEM**

Figure 8: The ablation results of the spatial-aware OHEM mechanism. The reconstructed HDR/WCG images are placed in the bottom-left corners of error maps.

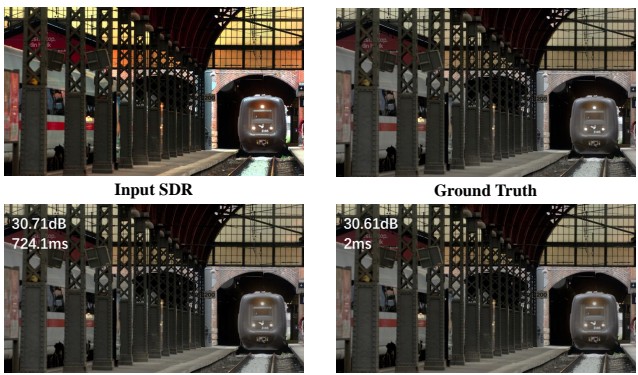

**Input SDR** · **Ground Truth**

**FMNet** · **Image-wise iTM Distillation**

Figure 9: The results of iTM acceleration application. Our framework could achieve comparable performance with existing methods [42] in significantly less time.

strategy, GamutMLP pre-training strategy, and our meta-training strategy. As shown in Table 4, our meta-learning pre-training strategy significantly improves the performance of the reconstruction. The pre-training strategy in GamutMLP [16] is similar to ours, but lacks the adaptation step before outer update, leading to inferior performance. The generalization evaluation results, shown in Table 3, also reveal the effectiveness of the meta-training strategy. **The effectiveness of the spatial-aware OHEM mechanism.** We evaluate the performance of the spatial-aware OHEM mechanism and the uniform sample strategy in our framework. The results, shown in Table 4, reveal that our spatial-aware OHEM mechanism efficiently improves the performance across all pre-training strategies. Additionally, we visualize error maps for different combinations in Fig. 8. It can be observed that the utilization of the spatial-aware OHEM mechanism yields superior reconstruction results and overall exhibits greater uniformity. Particularly, the challenging reconstruction region within the red frame further highlights the effectiveness of the spatial-aware OHEM sample mechanism.

## 4.6 Application: Image-wise iTM Distillation

There exists a considerable amount of SDR content that the ground truth HDR/WCG counterparts are not available. In such scenarios, we show that our framework is capable of working with existing iTM methods for the acceleration of HDR/WCG prediction by "distilling" them from a domain-wise model to an image-wise one. Specifically, we replace the ground truth HDR/WCG image in the transfer stage with results reconstructed by existing domain-wise iTM methods. Fig. 9 presents an example of the image-wise distillation results, demonstrating that our framework achieves comparable performance with significantly less latency.

## 5 CONCLUSION

In this work, we propose a meta-transfer learning framework for HDR/WCG media transmission by embedding an image-wise iTM model as metadata in the SDR version. To improve the generalization and adaptation capacity of our framework, we introduce a meta-learning strategy and a spatial-aware online mining mechanism. Through comprehensive experiments, our framework demonstrates significant superiority over existing solutions in terms of both performance and efficiency. Furthermore, our framework also generalizes well to different TM operators and can be utilized for iTM acceleration. Our future work includes extending our framework to the video iTM task, which has higher requirements for inter-frame coherence as well as latency.

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
