# OpenReview forum: "MLP Embedded Inverse Tone Mapping"
_acmmm.org/ACMMM/2024/Conference — MM2024 Poster_

### Official Review · Reviewer_L1w6 · 2024-05-05

**Rating:** 4
**Confidence:** 3

**Summary:**

This paper proposes a HDR format which is upconverted from SDR using metadata as MLP. The MLP is trained via meta learning to build a base network, and fine tune according to the content via OHEM. The authors show the experimental results both subjectively and objectively.

**Strengths:**

With a given set of TM operators to map down HDR to SDR, using meta learning is reasonable since most existing TM operators rule based, which can be reverse model back to HDR via MLP. Fine tuning for each HDR/SDR pair based on the common trained MLP should reduce the training time.

**Limitations:**

(1) It is unclear why transmitting HDR image needs 20MB and SDR just needs 2MB and based on which metric. Though it does not affect the proposed framework, having those numbers as motivations make this considered scenario less convincing. A well tuned HDR video codec can be as efficient (same bit rate) as SDR video codec with little metadata overhead in industrial practice.
(2) Adding coordinate information into MLP is supposed to handle the local tone mapping. Because without coordinates as input, with only RGB will make only global tone mapping. Local tone mapping are often content dependent. It will be great to explain why the meta learning can learn this content dependent local tone mapping. In addition, it will be great to have experiments to demonstrate the need of using coordinates as input to MLP.
(3) The experiments test Youtube TM.  It will be great to test other TM operators, too, to see how effective the proposed solution can be used.

**Suitability:**

2

---

### Official Review · Reviewer_KMM2 · 2024-05-17

**Rating:** 4
**Confidence:** 3

**Summary:**

The paper proposed a meta-transfer learning framework for practical HDR/WCG media transmission by embedding image-wise metadata into their SDR counterparts for later iTM reconstruction. A meta-learning strategy and a spatial-aware online mining mechanism is adopted to train a lightweight MLP domain-wise model. This model would be transmitted alongside the SDR image, facilitating the reconstruction of the original image on HDR/WCG displays.

**Strengths:**

Extensive experiments demonstrate that this method gains superior performance and minimal latency with negligible overhead.

**Limitations:**

1. It is suggested to unify the loss name in Algo.1 and Algo.2, using L1/L2 pair or MAE/MSE pair.
2. In Table 2, it is suggested to add FLOPs of each method to more intuitively show the calculation efficiency of all methods.
3. It is suggested that the authors also highlight the best experimental results by boldness in Table 2-4, just like Table 1.
4. The capitalization of articles’ titles and the format of publishers (abbreviations / full names) should be consistent in the reference section.

**Suitability:**

2

---

### Official Review · Reviewer_KTWv · 2024-05-24

**Rating:** 5
**Confidence:** 3

**Summary:**

This paper proposes a meta-learning framework for HDR media transmission. In this framework an HDR is compressed into a SDR with some metadata. The metadata is an image-wise MLP which can convert the SDR image to HDR. The proposed method outperforms existing ones in quality and efficiency.

**Strengths:**

1.It’s a good idea to embed HDR metadata as an image-wise MLP. MLP is suitable to learn the SDR to HDR mapping. And it’s fast and has just a little overhead.

2.The meta-learning strategy and OHEM sampling used by the proposed method can improve the generalization and adaptation capacity of this method.

3. The proposed method outperforms existing ones in quality and efficiency.

**Limitations:**

1. In my opinion, the task of this paper is more like an HDR compression task than inverse tone mapping, the proposed method may need to be compared with some image compression methods.

2.line 690 says that this method could reconstruct the over-exposure region accurately. But the MLP is a pixel to pixel mapping. Why it has the ability to accurately reconstruct over-exposed regions?

3.The image-wise ITM distillation seems like it still requires original ITM methods to convert each input SDR image first. What is the application scenario of the distillation?

**Suitability:**

2

---

### Official Review · Reviewer_m7QV · 2024-05-25

**Rating:** 5
**Confidence:** 4

**Summary:**

The paper presents a novel inverse tone mapping method that utilizes the idea of embedding metadata in SDR images. This approach addresses the storage and bandwidth challenges of HDR images by including metadata in the SDR transmission, which is later used to reconstruct the HDR image. Although the article is well-written, employs a good evaluation methodology, and shows promising results, some improvements are needed, and some questions remain unanswered.

### General Comments:

- The approach's dependency on the tone-mapping operator used to generate the HDR-SDR data pairs for pre-training is not clearly addressed. The authors do not mention the tone-mapping operator at the beginning but do so later in Section 4. Given that their proposed method uses a spatial-aware mechanism to fine-tune the model, different TM operators might show varying levels of effectiveness. The authors should clarify this dependency and include it in their article.
- All experiments have been conducted on static images. While this is mentioned as future work, the authors should explain whether the current method is stable over time and if it is prone to temporal artifacts.
- Professional visual content often includes completely black or white frames, as well as fade-in and fade-out sequences. The authors should discuss the performance of their method in these scenarios.



### Detailed comments:
### Section 1

- **Line 168:** This is not a contribution but a feature of the proposed iTM. This sentence should be reformulated to express the contribution. For example, "We present an iTM that..., which exhibits superior performance..."

### Section 2

- **Line 176:** Authors should include the following references that also propose a global method based on image features:

  - Bist, C., Cozot, R., Madec, G., & Ducloux, X. (2017). Tone expansion using lighting style aesthetics. Computers & Graphics, 62, 77-86.
  - Luzardo, G., Aelterman, J., Luong, H., Philips, W., Ochoa, D., & Rousseaux, S. (2018, June). Fully-automatic inverse tone mapping preserving the content creator’s artistic intentions. In 2018 Picture Coding Symposium (PCS) (pp. 199-203). IEEE.

### Section 3

- **Figure 2:** Please change the light-blue and light-green colors on the texts in the picture, as they are very difficult to read. It would be worse if the paper was printed.

- **Line 327:** What TM operator/method is used? Is it YouTube TM as defined in Section 4?

- **Line 337:** "What’s more, we adopt" -> "Additionally, we adopt"

### Section 4

- **Table 1:** The values reported by the authors correspond to the average between all the tested images. This should be mentioned by the authors.

- **Line 564:** HDR-VDP-3 works on the color domain as rgb-bt.2020.

- **Line 637:** "What’s more" -> "In addition", "Furthermore", etc.

### Section 4.6

- The application mentioned by the author is not clear. This section should be improved.

**Strengths:**

- **Fine-Tuning Strategy:** The strategy to fine-tune the model on hard samples using a spatial-aware OHEM mechanism is commendable.
- **Evaluation Strategy:** The use of a perceptual encoder for objective evaluation is a strong aspect of the paper.

**Limitations:**

- **Missing References:** There are a couple of references that should be included.

**Suitability:**

3

---

### Meta-Review · Area_Chair_Dx6i · 2024-06-30

**Recommendation:** Accept (Poster)
**Confidence:** 5

**Metareview:**

All reviewers are agreed to accept the paper.